# Generation and Characterization of a Human-Derived and Induced Pluripotent Stem Cell (iPSC) Line from an Alzheimer’s Disease Patient with Neuropsychiatric Symptoms

**DOI:** 10.3390/biomedicines11123313

**Published:** 2023-12-15

**Authors:** Ram Sagar, Cristina Zivko, Ariadni Xydia, David C. Weisman, Constantine G. Lyketsos, Vasiliki Mahairaki

**Affiliations:** 1Department of Genetic Medicine, Johns Hopkins University School of Medicine, Baltimore, MD 21205, USA; rsagar2@jhmi.edu (R.S.); czivko1@jhmi.edu (C.Z.); axydia1@jhu.edu (A.X.); 2The Richman Family Precision Medicine Center of Excellence in Alzheimer’s Disease, Johns Hopkins University School of Medicine, Baltimore, MD 21287, USA; kostas@jhmi.edu; 3The Department of Psychiatry and Behavioral Sciences, Johns Hopkins Medicine and Johns Hopkins Bayview Medical Center, Baltimore, MD 21287, USA; 4Abington Neurologic Associates, Clinical Research Center, Abington, PA 19001, USA

**Keywords:** iPSC generation, Alzheimer’s disease, neuropsychiatric symptoms, agitation

## Abstract

Agitation is one of the most eminent characteristics of neuropsychiatric symptoms (NPS) affecting people living with Alzheimer’s and Dementia and has serious consequences for patients and caregivers. The current consensus is that agitation results, in part, from the disruption of ascending monoamine regulators of cortical circuits, especially the loss of serotonergic activity. It is believed that the first line of treatment for these conditions is selective serotonin reuptake inhibitors (SSRIs), but these are effective in only about 40% of patients. Person-specific biomarkers, for example, ones based on in vitro iPSC-derived models of serotonin activity, which predict who with Agitation responds to an SSRI, are a major clinical priority. Here, we report the generation of human-induced pluripotent stem cells (iPSCs) from a 74-year-old AD patient, the homozygous APOE ε4/ε4 carrier, who developed Agitation. His iPSCs were reprogrammed from peripheral blood mononuclear cells (PBMCs) using the transient expression of pluripotency genes. These display typical iPSC characteristics that are karyotypically normal and attain the capacity to differentiate into three germ layers. The newly patient-derived iPSC line offers a unique resource to investigate the underlying mechanisms associated with neuropsychiatric symptom progression in AD.

## 1. Introduction

The neuropsychiatric symptoms (NPS), specifically agitation and psychosis, are considered the universal core symptoms of Alzheimer’s disease (AD) that occur in the preclinical, early clinical, as well as advanced clinical (dementia) phases of the disease [1]. These symptoms have significant adverse effects on patients and caregivers, including accelerated disease progression, significant disability, worse quality of life, and accelerated mortality [1,2,3]. A consensus group proposed that disruptions in the following types of circuits underlie NPS: frontal subcortical, cortical—cortical, and ascending monoamine “regulators” of cortical circuits [2].

Damage to the monoamine-producing nuclei, including dorsal raphe, is evident pathologically early in AD [3]. Clinical pathologic correlations consistently link NPS with dorsal raphe degeneration [4,5,6,7,8,9,10,11]. Available (modestly) effective therapies for psychosis and agitation are medications that target the dopamine and serotonin systems, such as atypical antipsychotics, SSRIs, or psychostimulants [3]. Recent evidence suggests that only subgroups of patients benefit from such medications (e.g., the agitation subgroup to the SSRI escitalopram [12]), raising questions as to how benefitting patients differs in the functioning of their mono-amine systems. There is also significant clinical variability in the timing and type of NPS emergence at different AD stages [1,2,3].

As NPSs are inherently human symptomatic expressions, animal models have not been successfully developed. While animal models have contributed to understanding the interactions between monoamine systems and the amyloid cascade in transgenic mice, they have not been helpful in understanding NPS mechanisms. Therefore, the opportunity to acquire knowledge at the individual level of the efficiency of monoamine systems as a personalized medicine approach would be a major advance.

Human iPSCs provide unprecedented opportunities for regenerative medicine applications. They can be routinely derived from somatic cells in unlimited supplies and can be re-differentiated with methodologies that optimize the safety of pluripotent cells and their progeny [13,14]. Human iPSCs have the capability to differentiate into all functional cell types; they can also capture the unique genomic information of the donors, thus serving as an excellent source to understand molecular pathogenesis and develop targeted treatments [15,16].

In this case study, we generated a human iPSC line from a 74-year-old Master’s-prepared male with AD enrolled in a clinical trial evaluating the safety and efficacy of escitalopram, an SSRI, for the treatment of Agitation in people with AD (ClinicalTrials.gov-Identifier: NCT03108846). He was first diagnosed with dementia 3 years earlier and, at the time of study enrollment, had a Mini-Mental State score of 12/30, with moderate functional dependency (50 on the ADCS-ADL scale). He first developed Agitation about a year before, and by study, entry was moderate in severity (22 on the Neuropsychiatric Inventory Clinician Rating). He had not improved on other treatments for Agitation. This patient’s derived iPSC line offers a unique opportunity to shed light on NPS mechanisms. Generating iPSCs from more AD patients with NPS could also allow the differences in therapeutic responses to be studied, thus paving the way for the development of patient-specific drug-screening platforms.

## 2. Materials and Methods

### 2.1. Human Induced Pluripotent Stem Cell Generation 

The SAN013 iPSC line was generated using episomal vectors under feeder-free/xeno-free culture conditions, as described in previous studies [17,18,19]. Briefly, after informed consent was provided, blood was collected from the patient using a study collaborator (DCW) through the ongoing S-CitAD clinical trial with oversight by the Johns Hopkins Institutional Review Board (IRB) internal review board. Peripheral blood mononuclear cells (PBMCs) were isolated at the Johns Hopkins Core facility and then cultured to enrich erythroblasts. Afterward, the erythroblasts were reprogrammed via nucleofection, which involved the transient expression of MOS, MMK, and GBX episomal vectors (Addgene, Watertown, MA, USA) through the 4D Nucleofector (Lonza, Basel, Switzerland), as previously published [17]. The cells were then transferred to plates coated with vitronectin (Gibco, Grand Island, NY, USA) and cultured in a DMEM medium (Gibco, Grand Island, NY, USA) containing 10% of the fetal bovine serum (FBS). After one day, the serum containing the medium was replaced with 50% of the Essential 8 (E8) medium (Gibco, Grand Island, NY, USA) supplemented with sodium butyrate (NaB) (Millipore Sigma, St. Louis, MO, USA) every other day until colony formation. At days 14–19 post-transfection, the newly developed colonies were pooled, and the TRA-1-60 positive pluripotent cells were isolated via positive selection with anti-TRA-1-60 MicroBead-kit (Miltenyi Biotec, Auburn, CA, USA). The positive cells were cultured (250,000 cells/well) in an E8 medium supplemented with a 10 uM rock inhibitor (Y-27632) (STEMCELL Technologies, Cambridge, MA, USA) on vitronectin-coated tissue-culture plates (250,000 cells/well) and the E8 medium was changed daily [18,19].

### 2.2. Flow Cytometric Analysis

The flow cytometric analysis was conducted according to previously described study [17]. Briefly, the human iPSCs were dissociated with TrypLE (Gibco, Grand Island, NY, USA) into a single-cell suspension in PBS and centrifuged. They were then resuspended with the BD FACS-staining buffer (Thermo Fischer, Carlsbad, CA, USA) upon resuspension in this buffer, and the cells were labeled with the anti-human TRA-1-60 antibody (Millipore Sigma, St. Louis, MO, USA). The anti-mouse IgM control, PE, conjugated (R&D Systems, Minneapolis, MN, USA) was used as a control. Flow cytometric analysis was conducted on a BD LSR Fortessa analyzer (BD Biosciences, San Jose, CA, USA), and FlowJo software (v10.8.1) was used for data analysis.

### 2.3. APOE Genotyping

The genotyping of the induced pluripotent stem cells (iPSCs) was performed using the polymerase chain reaction-restriction fragment length polymorphism (PCR-RFLP) method. Genomic DNA was extracted from the iPSCs using the DNeasy kit (Qiagen, Germantown, MA, USA). To confirm the genotypes, Sanger sequencing was conducted. Gene-specific primers were used for PCR amplification (Table 1). The ABI Prism 3730 XL genetic analyzer (Applied Biosystems, Waltham, MA USA) was used for gene sequencing and the analysis of the amplified PCR products.

### 2.4. Immunocytochemistry Staining for Pluripotency Markers

Cultured human iPSCs in 12-well plates were washed in PBS and fixed in freshly prepared 4% (*v*/*v*) paraformaldehyde (PFA) in PBS (pH 7.4) for 20 min. For the staining of nuclear antigens, 0.1% Triton X-100 (*v*/*v* in PBS) was used for permeabilization. The cells were blocked in 10% goat serum for 1 h at 4 °C, and they were subsequently stained with primary antibodies (anti-human TRA-1-60, NANOG, and OCT4) at 4 °C overnight. The next day, cells were washed with PBS, they were incubated with the appropriate secondary antibodies (see Table 1) for 1 h, washed 3 times with PBS, and counterstained with DAPI.

### 2.5. In Vitro Trilineage Differentiation

In vitro trilineage differentiation was conducted using the StemMACS ^TM^ Trilineage Differentiation Kit, Human (Miltenyi Biotec, Auburn, CA, USA) as per the manufacturer’s instructions. Briefly, iPSCs were seeded onto a Matrigel-coated 12-well plate after single-cell dissociation and cultured in the trilineage differentiation-specific medium until day 7. About 250,000 cells/well were seeded for endoderm differentiation, 200,000 cells/well for ectoderm, and 150,000 cells/well for mesoderm. The cells were fixed with 4% PFA and stained for Alpha-fetoprotein (endoderm), smooth muscle actin (mesoderm), and β-tubulin III (ectoderm) (antibodies listed in Table 1).

### 2.6. Karyotyping Analysis

Human iPSCs were submitted to the Johns Hopkins Cytogenetics Core Facility Center for G-band karyotyping when the cells were at 70–80% confluency. About 20 metaphases were counted, and the structural evaluation of G-banding was performed at a 450–500 band resolution.

### 2.7. Short Tandem Repeats (STRs) Analysis

DNA from both parental PBMCs and the generated iPSC line were analyzed at the Johns Hopkins Genetic Resources Core Facility. The polymerase chain reaction (PCR) amplification of ten short tandem repeat (STR) loci was performed with a Promega GenePrint 10 Kit. Amelogenin (*AMEL*), a gender-determining marker, was included in the analysis. The ABI Prism^®^ 3730Xl genetic analyzer was used for the electrophoresis of the PCR product, and GeneMapper^®^ v 4.0 software (Applied Biosystems, Waltham, MA, USA) was used for data analysis.

### 2.8. Mycoplasma Detection 

The hiPSC culture medium was collected and analyzed for mycoplasma contamination using the PCR-based MycoDect™ kit (Greiner Bio-One, Monroe, NC, USA).

### 2.9. Lentiviral Transduction and Neuronal Differentiation

We performed the neuronal differentiation and lentivirus transduction to obtain the high yield of glutamatergic excitatory neurons for this newly generated hiPSC using rtTA and NgN2-expressing lentiviruses (Cellomics Technology, Arbutus, MD, USA), as per the previously published protocol [20]. The established hiPSC was transduced with NgN2 and rtTA-expressing lentiviruses supplemented with polybrene when they reached the confluence for up to 50% and were incubated for 6 h at 37 °C. After the recommended time of incubation with lentivirus, the medium was replaced with fresh E8, and incubation was conducted to reach the confluency of 90%. Then, the transduced cells were passaged (250,000 cells/well) on a vitronectin coated plate in the E8 medium supplemented with a rock inhibitor. The medium was replaced with an induction medium (iN-N2) and supplemented with doxycycline after 48 h of the culture in E8 media to start the differentiation procedure. The next day, puromycin was added to the induction media for the selection of transduced cells for 24 h. The surviving cells were then passed (500,000 cells/well) onto the Matrigel-coated plate in a neurobasal medium for neuronal differentiation. The protocol was followed, as described in our previously published paper, and at day 30, mature neurons were collected to perform gene expression using qPCR [20].

### 2.10. Immunocytochemistry Staining for Neuronal Markers

We used the same protocol for the immunostaining of differentiated neurons on day 15 and used the primary antibody (anti-human MAP 2 and TUJ1) as described in the methodology Section 2.4.

### 2.11. Quantitative RT-PCR

We collected the hiPSC of the SAN013 cell line before the start of neuronal differentiation and, on day 30, collected neuronal cell pellets to extract the total RNA using Quick-RNA Miniprep Kit (Zymo Research, Irvine, CA, USA) as per the manufacturer’s guidelines. Then, the cDNA was synthesized using High-Capacity RNA and the cNDA Kit (Applied Biosystems, Waltham, MA, USA). Lastly, we performed the gene expressions for the specific PCR primers listed in Table 2. A total of 3 µL of cDNA was used as a template for every single reaction mixture in real-time quantitative PCR analysis using the SsoAdvanced Universal SYBER green supermix (Biorad, Hercules, CA, USA). The housekeeping gene (*GAPDH*) was used as a control gene to normalize the relative expression of pluripotency markers (TRA-1-60 and NANOG) in the hiPSC. The fold change in gene expression for target genes (neuronal markers) was analyzed for the gene expression of iPSC.

## 3. Results and Discussion

We generated and characterized iPSCs from the peripheral blood mononuclear cells (PBMCs) of a 75-year-old patient (Figure 1). PBMCs were reprogrammed via the transfection of three episomal vectors expressing human OCT4/SOX2 (MOS), C-MYC/KLF4 (MMK), and BCL-XL (GBX) genes which increased the reprogramming efficiency up to 50-fold [20,21]. The generated iPSCs formed colonies with well-defined borders and a high nuclear/cytoplasmic ratio resembling the morphology of human embryonic stem cells, as shown by light microscopy (Figure 2A). Pluripotency was evaluated both by immunocytochemistry for pluripotency markers such as OCT4, TRA-1-60, and NANOG markers (Figure 2B), as well as flow cytometry for TRA-1-60 positive cells (Figure 2C). The pluripotency of newly generated hiPSC was also confirmed with the gene expression of pluripotency markers (*TRA-1-60 and NANOG*) using RT-PCR (Figure 3A). Using Sanger sequencing, we detected the presence of two *APOE ε4* alleles in the iPSCs (Figure 2D). The karyotype analysis of about 20 metaphases indicated that the cells were 46, XY (Figure 2E). To assess pluripotency, we performed tri-lineage differentiation via embryoid body formation (EB), confirming the generation of ectoderm (β-tubulin III), mesoderm(α-SMA), and endoderm (AFP) (Figure 2F). Short tandem repeat (STR) based analysis showed 100% identity between the parental (PBMC) and the cultured (iPSC) line (Figure 2G). Mycoplasma testing of the SAN013 iPSC line showed negative results. Then, the potential to differentiate the glutamatergic excitatory neuronal identity was evaluated with different neuronal markers (e.g., *MAP2, VGLUT1, NMDAR1*, and *CTIP2*) using RT-PCR (Figure 2B), morphology via light microscopy (Figure 2C) and neuronal identity via the immunocytochemistry of neurons specific markers such as TUJ1 and MAP2 (Figure 2D). This patient’s derived iPSC line offers a unique resource to study the mechanisms associated with neuropsychiatric symptom progression in AD.

Neuropsychiatric symptoms (NPS) encompass a wide range of cognitive, emotional, and behavioral disturbances observed in AD patients, arising from early clinical to advanced stages of the disease [21,22,23]. These are among the foremost contributors to the disease burden affecting both patients and caregivers. The previous literature provides strong evidence that NPS accelerates the progression of cognitive and functional decline in patients with AD [24,25]. Studies have found that Agitation occurs in around 60% of patients with AD and is associated with depression, delusions, and other NPSs [26,27]. The symptoms of agitation and aggression are more common in patients with Alzheimer’s disease who are homozygous carriers of *APOE4* alleles [28,29,30,31]. Few studies assess the association of *APOE* genotypes. Understanding the complex interplay between the *APOE* genotype and NPS holds promise for improving diagnostic and therapeutic approaches.

This is a case report detailing the generation and characterization of a distinct iPSC line derived from an Alzheimer’s disease (AD) patient with neuropsychiatric symptoms (NPS). Patient-derived iPSCs offer the possibility for novel, disease-relevant in vitro models that can be helpful in understanding neuropathological mechanisms using a precision medicine approach [32]. Our work establishes a human iPSC line with specific clinical manifestations of NPS from a genotypically defined AD patient carrying *APOE ε4/ε4*. This serves as a fundamental step for future in vitro studies aimed at decoding the correlation between the molecular pathology behind *APOE ε4/ε4* in AD and the development of NPS.

Reprogrammed iPSCs from specific subsets of patients, such as the one reported herein, hold great potential for disease modeling, which can be used to study poorly understood pathophysiological mechanisms of AD and NPS. An iPSC-based model of NPS in AD can help to illuminate the impact of high-risk genetic variants and their relevance on physiological neuronal functioning. In addition to these broader genetic studies, cells derived from patients with specific symptomatology can be used for a variety of in vitro experiments. Single-cell and co-culture systems of iPSCs-derived cells are being increasingly explored for analogous purposes, including three-dimensional organoid cultures of specialized brain regions [33,34]. This increasing complexity can help elucidate disease mechanisms that emerge from the interplay between multiple cell types and across tissues. Studies have shown a link between the manifestation of NPS in AD and disruptions in the serotonergic neurons in the raphe-nuclei, which could be recapitulated using iPSC-derived serotonergic cells and/or organoids [35,36,37,38].

At the same time, patient-specific cells and tissues of interest can be used as biomarkers to screen commercially available and/or novel therapeutic agents in vitro before exposing people to drugs they may or may not benefit from. This is becoming a topic of particular interest as of December 2022 since newly signed legislation in the US established that the Federal Drug Administration (FDA) no longer requires animal testing before human trials for drug approval [39].

Our personalized medicine approach could eventually be applied to study differences in therapeutic responses, starting with iPSCs as clinically relevant in vitro models. We believe that the iPSC line reported here could be highly useful in studying the neurobiology of NPS, such as agitation, the effect of the *APOE ε4/ε4* genotype, and responsiveness (or lack thereof) to pharmacological treatment options.

## Figures and Tables

**Figure 1 biomedicines-11-03313-f001:**
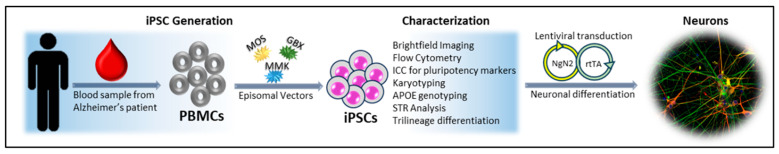
Schematic study diagram.

**Figure 2 biomedicines-11-03313-f002:**
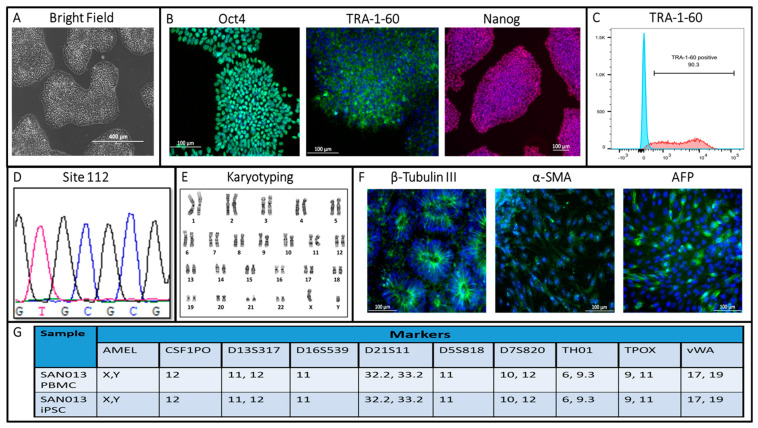
Characterization of a novel iPSC line from an AD patient with NPS. (**A**) iPSC morphology using bright field microscopy (scale bar: 400 µm). (**B**) Immunostaining using pluripotency markers OCT4, TRA-1-60, and Nanog (scale bar: 100 µm). (**C**) Purity of human TRA-1-60^+^ cells as seen via flow cytometry. (**D**) Genotyping using DNA sequencing. (**E**) Chromosome Analysis using Karyotyping. (**F**) Immunostaining for Trilineage differentiation capacity using Endoderm (AFP), Mesoderm (α-SMA), and Ectoderm (β-tubulin III) markers (scale bar: 100 µm). (**G**) Short tandem repeat (STR) profiling analysis between newly generated iPSC line and parental PBMC.

**Figure 3 biomedicines-11-03313-f003:**
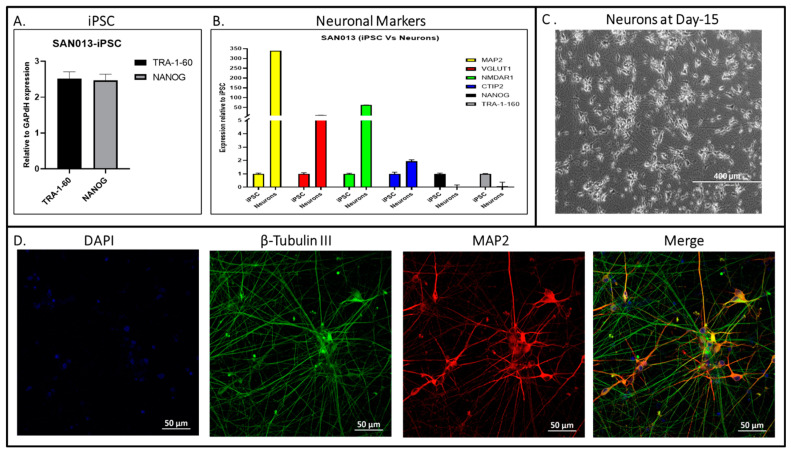
Characterization of neurons derived from the hiPSC (SAN013) line from AD patients with NPS. (**A**) Gene expression using qPCR for the pluripotency of iPSC (*TRA-1-60* and *NANOG*). (**B**) Gene expression of neuronal markers (*MAP2, VGLUT, NMDAR1,* and *CTIP2*) to show neuronal identity. (**C**) Neuronal morphology using bright field microscopy (scale bar: 400 µm). (**D**) Representative immunostaining of β-Tubulin III (TUJ1) and MAP2-positive neurons on day 15 (scale bar: 50 µm).

**Table 1 biomedicines-11-03313-t001:** List of antibodies and primers.

	Antibody and Host Species	Dilution	Manufacturer
Pluripotency markers	TRA-1-60, Mouse IgM	1:300	EMD Millipore, MAB4360, RRID:AB_3548341
	OCT4, Rabbit IgG	1:200	Santa Cruz Biotechnology, sc-9081, RRID:AB_E1011
	NANOG, Mouse IgG	1:200	BD Biosciences, 560482, RRID:AB_8240787
Differentiation markers	β-tubulin III, Mouse IgG	1:500	Biolegend, 801201, RRID:AB_B205808
	AFP, Mouse IgG	1:100	Thermo Fisher Scientific, MA5-14666, RRID:AB_VI308040
	α-SMA, Mouse IgG	1:50	EMD Millipore, CBL171, RRID:AB_2223166
	MAP2 (D5G1), Rabbit IgG	1:400	Cell Signaling, 8707, D5G1, RRID:AB_2722660
Secondary antibodies	Alexa Fluor 555 goat anti-mouse IgM	1:500	Thermo Fisher Scientific, A-21426, RRID:AB_2128995
	Alexa Fluor 488 goat anti-mouse IgG	1:500	Thermo Fisher Scientific, A-11001, RRID:AB_1907294
	Alexa Fluor 555 goat anti-Rabbit IgG	1:500	Thermo Fisher Scientific, A-21428, RRID:AB_ 2535849
	Alexa Fluor 488 goat anti-rabbit IgG	1:500	Thermo Fisher Scientific, A-11034, RRID:AB_1885241
Primers for *APOE* Genotyping	**Forward/Reverse primer (5′–3′)** F: GGCACGGCTGTCCAAGGA; R: GCCCCGGCCTGGTACAC		

**Table 2 biomedicines-11-03313-t002:** List of primers for real-time quantitative PCR analysis.

Markers	Forward Primer	Reverse Primer
TRA-1-60	ACAGGAAACACCCTCTGTGC	GAAGGTGGCTTTGACTGCTC
NANOG	ACAACTGGCCGAAGAATAGCA	GGTTCCCAGTCGGGTTCAC
MAP2	TC GAGGCAATGACCTTACC	GTGGTAGGCTCTTGGTCTTT
VGLUT1	CACCATGGAGTTCCGCC	CACTCAGCTCCAGCGTCTC
NMDAR1	ATCTACTCGGACAAGAGCATCC	AGCTCTTTCGCCTCCATCAG
CTIP2	CAGAGCAGCAAGCTCACG	GGTGCTGTAGACGCTGAA GG

## Data Availability

The data presented in this study are available on request from the corresponding author.

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
