# Peer review of "Generation and Characterization of a Human-Derived and Induced Pluripotent Stem Cell (iPSC) Line from an Alzheimer’s Disease Patient with Neuropsychiatric Symptoms"

_biomedicines, 2023, doi:10.3390/biomedicines11123313_

Round 1
Reviewer 1 Report
Comments and Suggestions for Authors
In this manuscript, the author generated an iPSC line from a 74-year-old patient with Alzheimer’s disease carrying the APOE ε4/ ε4 genotype. The characterization of this iPSC line is almost characterized regarding the pluripotency characteristics of the cells: expression of pluripotency markers by immunofluorescence staining and FACS analysis, differentiation into derivatives of the three germ layers (endoderm, ectoderm, and mesoderm). The iPSC line has a normal karyotype and characterized by APOE genotyping.
Result part :
Figure 1A: some of the colonies are differentiating in the center. The shape of the border of the colonies is also indicating of differentiation.
Figure 1B : the doble staining OCT4 and DAPI is not convincing as only the cells at the borders are expressing OCT4.
Figure 1B : the doble staining TRA1-60 and DAPI is not convincing as only ~half of the cells are expressing TRA1-60.
In general, such staining should be analysed by a quantitative analysis eg. an histogram showing the percentage of cells expressing the pluripotency marker.
Another possibility is to show quantitative qPCR analysis of the expression of pluripotency genes such as OCT4, NANOG, SOX2 or others by comparison with an established human embryonic stem cells such as H1-ESCs, H9-ESCs or others.
The results of only one iPSC line are not sufficient. Two iPSC lines from the same patient should be generated and described (except if iPSCs are reprogrammed from at least 2 affected patients).
Discussion part :
Line 179-180 : “Here we have generated an iPSC line for the very first time from an AD patient with 179 NPS carrying the APOE ε4/ε4 genotype”.
This is not true. Numerous papers reporting the generation of iPSCs from patient with Alzheimer’s disease carrying the APOE ε4/ ε4 genotype have been published in Stem Cell Research journal. None of these articles is cited in references by the authors.
Establishment of SIAISi021-A, an induced pluripotent stem cell (iPSC) line from a 71-year-old Chinese Han male with Alzheimer’s Disease (AD) having two copies of APOE4/4 allele
Stem Cell Research14 July 2022...
1. Yanfei Ding
2. Haijuan Chen
3. Yulei Deng
Generation of a set of induced pluripotent stem cell lines from two Alzheimer disease patients carrying APOE4 (MLUi007-J; MLUi008-A) and healthy old donors carrying APOE3 (MLUi009-A; MLUi010-B) to study APOE in aging and disease
Stem Cell Research15 March 2023...
1. Matthias Jung
2. Carla Hartmann
3. Dan Rujescu
Generation of an induced pluripotent stem cell line (SIAISi009-A) from a 60-year-old Chinese Han female with mild cognitive impairment (MCI) having two copies of APOE4/4 allele
Stem Cell Research26 January 2021...
1. Jinghui Guo
2. Weihao Di
3. Ying Wang
Establishment of SIAISi001-A, an induced pluripotent stem cell (iPSC) line from 66-year old mild cognitive impairment (MCI) with two copies of APOE4 gene
Stem Cell Research25 January 2020...
1. YI Yan
2. Yinghui Qiu
3. Yulei Deng
Generation and characterization of human induced pluripotent stem cell (hiPSC) lines from an Alzheimer's disease (ASUi003-A) and non-demented control (ASUi004-A) patient homozygous for the Apolipoprotein e4 (APOE4) risk variant
Stem Cell ResearchDecember 2017...
1. Nicholas Brookhouser
2. Ping Zhang
3. David A. Brafman
Generation and characterization of human induced pluripotent stem cell (hiPSC) lines from an Alzheimer's disease (ASUi001-A) and non-demented control (ASUi002-A) patient homozygous for the Apolipoprotein e4 (APOE4) risk variant
Stem Cell ResearchOctober 2017...
1. Nicholas Brookhouser
2. Ping Zhang
3. David A. Brafman
Generation and characterization of two human induced pluripotent stem cell (hiPSC) lines homozygous for the Apolipoprotein e4 (APOE4) risk variant—Alzheimer's disease (ASUi005-A) and healthy non-demented control (ASUi006-A)
Stem Cell ResearchOctober 2018...
1. Nicholas Brookhouser
2. Ping Zhang
3. David A. Brafman
Human induced pluripotent stem cell (iPSC) line (HEBHMUi014-A) derived from a patient with Alzheimer's disease
Stem Cell Research6 May 2023...
1. Mingjing Zhang
2. Mei Yu
3. Jun Ma
Establishment of SIAISi002-A, an induced pluripotent stem cell (iPSC) line from 39-year old healthy female donor with a family history of Alzheimer's disease (AD) and two copies of APOE4 gene
Stem Cell Research22 April 2020...
1. Yi Yan
2. Yinghui Qiu
3. Yulei Deng
Episomal plasmid-based generation of an iPSC line from a 79-year-old individual carrying the APOE4/4 genotype: i11001
Stem Cell ResearchNovember 2016...
1. Shadaan Zulfiqar
2. Barbara Fritz
3. Katja Nieweg
Episomal plasmid-based generation of an iPSC line from an 83-year-old individual carrying the APOE4/4 genotype: i10984
Stem Cell ResearchNovember 2016...
1. Shadaan Zulfiqar
2. Barbara Fritz
3. Katja Nieweg
Integration-free induced pluripotent stem cell line derived from a 62-years-old male donor with APOE-epsilon4/epsilon4 alleles
Stem Cell Research12 March 2022...
1. Ruiyun Guo
2. Xin Liu
3. Huixian Cui
Blood-derived integration-free induced pluripotent stem cells (iPSCs) from one 53-years-old male donor with APOE-ε4/ε4 genotype
Stem Cell Research27 June 2021...
1. Xiaowei Ma
2. Jun Ma
3. Huixian Cui
Induced pluripotent stem cells derived from one 70-years-old male donor with the APOE-ε4/ε4 alleles
Stem Cell Research13 May 2021...
1. Jin Wang
2. Xin Liu
3. Huixian Cui
The authors point to neuropsychiatric symptoms such as agitation and psychosis and the important role of serotonin pathway in those symptoms in the abstract, introduction and discussion parts. However, no investigations regarding these aspects have been done. Does this iPSC line recapitulate the phenotype? Does the author find the involvement of serotonin pathway in these iPSC line.
My general analysis of the manuscript doesn’t represent a real novelty (numerous papers have been published with iPSCs carrying the APOE ε4/ ε4 genotype) and there are no further investigations regarding the phenotype associated with the patient used for the reprogramming. All these concerns make this manuscript not acceptable for publication in Biomedicines.
Reviewer 2 Report
Comments and Suggestions for Authors
Here, the authors report in detail their method to generate human induced pluripotent stem cells (iPSCs) from a 74-year-old Alzheimer’s disease (AD) patient who was a homozygous APOE ε4/ε4 carrier,. His dementia was associated with severe agitation. The iPSCs were reprogrammed from peripheral blood mononuclear cells (PBMCs) by engineering transient expression of pluripotency genes and the iPSCs were shown to be karyotypically normal and they differentiated into cells of the three germ layers. It is concluded that this patient’s derived iPSC line offers a resource to study the mechanisms associated with neuropsychiatric symptom development and progression in AD.
The generation of the pluripotent iPSCs from this patient’s bone marrow is described in great detail which will be of value to other investigators wishing to do similar or access the authors’ cell line. I had, however, several issues with this report:
First, we are not told how the diagnosis of AD was confirmed. Did the subject have amyloid brain PET or CSF studies to support the diagnosis? Dementia with Lewy bodies is more commonly associated with neuropsychiatric symptoms than AD. Was DLB excluded as a diagnosis?
Second, is there an intention to also generate iPSCs from AD cases without NPS as a control?
Third, while the investigtors have generated pluripotent iPSCs from their patient, they don’t discuss in any detail how these could be used to determine the underlying cause of the agitation or other NPS present in this AD case. Will whole genome screening or a screen for mutations in selected candidate genes be performed? Will monoamine and their receptor and transporer levels be measured? As it stands, the report simply describes the process of how to induce pluripotent iPSCs from bone marrow cells which is already established.
Reviewer 3 Report
Comments and Suggestions for Authors
In this paper, the authors described the generation of an iPS cell line from the PBMC of a patient with Alzheimer’s disease. The generation of such cell lines is important because it will provide a better understanding of the disease, and allow optimizing the therapeutic strategy for the subgroups of the patients. Although the paper is interesting, some points should be considered before publication.
1. The authors should describe the steps of the generation of the iPS in detail because it is the first iPS that is generated from the PBMC of a patient with Alzheimer’s disease (Here we have generated an iPSC line for the very first time from an AD patient with NPS carrying the APOE ε4/ε4 genotype). They also can show the workflow using a schematic diagram.
2. When they characterized the iPS, like differentiation to mature cell types, did they notice any difference compared to iPS from healthy subjects? At least they can discuss this point. This will be important for their concept of personalized medicine.
3. The discussion section looks just like a repetition of the introduction. Since this is a short paper that describes a method, I suggest combining the result and discussion section.
4. The authors should compare the monoamine oxidase system of their iPS-derived neurons and the neurons derived from healthy iPS.
5. There are some typing errors in the manuscript.
Comments on the Quality of English LanguageThere are some typing errors in the manuscript.
Round 2
Reviewer 1 Report
Comments and Suggestions for Authors
In this manuscript, the author generated an iPSC line from a 74-year-old patient with Alzheimer’s disease carrying the APOE ε4/ ε4 genotype. The characterization of this iPSC line is almost characterized regarding the pluripotency characteristics of the cells: expression of pluripotency markers by immunofluorescence staining and FACS analysis, differentiation into derivatives of the three germ layers (endoderm, ectoderm, and mesoderm). The iPSC line has a normal karyotype and characterized by APOE genotyping.
Compared to the first round of revisions, the authors added data on the ability of this iPSC line to differentiate into neurons with an upregulation of neuronal genes (MAP2, VGLUT, 359 NMDAR1 and CTIP2) which completes the characterization of this iPSC line.
However, the second key issue remains: Does this iPSC line recapitulate the phenotype? Does the author find the involvement of serotonin pathway in these iPSC line? No results are presented in this manuscript regarding the presence or absence of a phenotype, only preliminary data regarding the serotoninergic pathway are presented in the authors’ comments but not in the manuscript.
My general analysis of the manuscript is the same as before this manuscript doesn’t represent a real novelty and there are no further investigations regarding the phenotype associated with the patient used for the reprogramming. All these concerns make this manuscript not acceptable for publication in Biomedicines.